# Phosphoethanolamine Transferases as Drug Discovery Targets for Therapeutic Treatment of Multi-Drug Resistant Pathogenic Gram-Negative Bacteria

**DOI:** 10.3390/antibiotics12091382

**Published:** 2023-08-29

**Authors:** Van C. Thai, Keith A. Stubbs, Mitali Sarkar-Tyson, Charlene M. Kahler

**Affiliations:** 1The Marshall Center for Infectious Diseases Research and Training, School of Biomedical Sciences, University of Western Australia, Crawley, WA 6009, Australia; vanchi.thai@research.uwa.edu.au (V.C.T.); mitali.sarkar-tyson@uwa.edu.au (M.S.-T.); 2School of Molecular Sciences, University of Western Australia, Crawley, WA 6009, Australia; keith.stubbs@uwa.edu.au

**Keywords:** phosphoethanolamine transferases, EptA, MCR, polymyxin resistance, Gram-negative bacteria

## Abstract

Antibiotic resistance caused by multidrug-resistant (MDR) bacteria is a major challenge to global public health. Polymyxins are increasingly being used as last-in-line antibiotics to treat MDR Gram-negative bacterial infections, but resistance development renders them ineffective for empirical therapy. The main mechanism that bacteria use to defend against polymyxins is to modify the lipid A headgroups of the outer membrane by adding phosphoethanolamine (PEA) moieties. In addition to lipid A modifying PEA transferases, Gram-negative bacteria possess PEA transferases that decorate proteins and glycans. This review provides a comprehensive overview of the function, structure, and mechanism of action of PEA transferases identified in pathogenic Gram-negative bacteria. It also summarizes the current drug development progress targeting this enzyme family, which could reverse antibiotic resistance to polymyxins to restore their utility in empiric therapy.

## 1. Introduction

Antimicrobial resistance (AMR) has been identified by the World Health Organization (WHO) as one of the most pressing threats to global public health [1]. The emergence of novel AMR mechanisms is leading to the mismanagement of commonly occurring infectious diseases, thereby prolonging transmission chains, and exacerbating morbidity and mortality rates [2]. The treatment of AMR infections is associated with significantly higher healthcare costs relative to the treatment of common infections, as delayed care in these patients may necessitate prolonged hospitalization, increase the risk of intensive care, and require diagnostics-driven point-of-care testing for AMR and non-standard therapeutic pathways [3]. In 2016, O’Neill et al. [4,5] reported that 700,000 people worldwide die annually due to antimicrobial-resistant infections, and this number is projected to reach 10 million cases by 2050, costing global healthcare systems nearly $8 trillion [4,5]. A recent systematic analysis showed that 4.95 million deaths in 2019 were associated with bacterial AMR, indicating that the forecasts by O’Neill et al. [4,5] are gradually becoming a reality.

In 2017, WHO published a list of antibiotic-resistant “priority pathogens,” including 12 bacterial species that require urgent research and development of novel antibiotics [6]. Infections caused by these species are generally more difficult, and sometimes impossible, to treat as they are resistant to most first-in-line antibiotics (e.g., β-lactams, aminoglycosides, and fluoroquinolones) [6]. The majority of pathogens in the WHO’s list are Gram-negative bacteria, including *Acinetobacter baumannii*, *Pseudomonas aeruginosa* (*P. aeruginosa*), *Enterobacteriaceae*, *Neisseria gonorrhoeae*, *Helicobacter pylori*, *Campylobacter* spp., *Salmonellae*, *Haemophilus influenzae*, and *Shigella* spp. [6]. Gram-negative bacteria exhibit higher antibiotic resistance than Gram-positive bacteria due to their outer membrane (OM) enclosing the cell wall, which impedes the permeability of antibiotics to reach their intended targets in the periplasm and cytosol [7].

Polymyxins, including polymyxin B (PxB) and colistin, are cationic antimicrobial peptides (CAMPs) that are primarily used as a last-in-line treatment of infections caused by multidrug-resistant (MDR) Gram-negative bacteria, particularly carbapenem-resistant *A. baumannii*, *P. aeruginosa*, and *Enterobacteriaceae* [8]. Polymyxins exhibit their antibacterial activity by targeting the bacterial OM and binding to the negatively charged phosphate groups of lipid A on lipopolysaccharides (LPS)/lipooligosaccharide (LOS) via the α, γ-diaminobutyric acid residue of the positively charged polymyxin [9]. This binding leads to increased permeability and the loss of bacterial membrane integrity, which ultimately results in bacterial cell lysis.

Unfortunately, bacteria have evolved numerous mechanisms to develop resistance to polymyxins, including the targeted modification of LPS/LOS, reduced production of LPS/LOS or lipid A, increased expression of efflux systems, altered expression of OM proteins, and increasing production of anionic capsular polysaccharide [10]. In pathogenic Gram-negative bacteria, the modification of lipid A headgroups on LPS/LOS with cationic residues (e.g., phosphoethanolamine (PEA) and 4-amino-4-deoxy-L-arabinose (L-Ara4N)) has been reported to be the primary mechanism of resistance by disrupting the interaction between polymyxin and lipid A to confer high-level polymyxin resistance (Figure 1a) [10,11].

The modification of lipid A headgroups with PEA occurs in the periplasm, where the PEA moiety is transferred from phosphatidylethanolamine (PE) to the 1 and/or 4′ position on Lipid A to form PPEA-Lipid A (Figure 1b) [12,13]. Unlike L-Ara4N modification, which is generally catalyzed by the chromosome-encoded enzymes, the decoration of PEA to the lipid A headgroups is catalyzed by the family of PEA transferases, which can be found on core genomes, genetic islands, or plasmids of many Gram-negative bacteria [14]. The following literature review endeavours to provide a systematic summary of the fundamental functions, structures, and mechanisms of action exhibited by PEA transferases that have been identified in pathogenic Gram-negative bacteria. Additionally, this review aims to outline the current progress in drug development targeting this family of enzymes.

## 2. PEA Transferases

PEA transferases are members of the alkaline phosphatase superfamily (IPR017850), which are primarily responsible for attaching PEA to lipid A headgroups of LPS/LOS in Gram-negative bacteria to confer CAMP resistance [15]. Recent evidence has revealed the existence of diverse PEA transferases in pathogenic Gram-negative bacteria, which facilitate PEA-dependent modifications on multiple sites of proteins and glycans [16,17]. Previously, Harper et al. [18] conducted a comprehensive phylogenetic analysis of known and predicted PEA transferases, identifying five distinct families based on their substrate specificity. These families include PEA transferases that are specific to the lipid A headgroups, the O-3 position of HepII, the O-6 position of HepII, Gal residues, and Kdo in the inner core of LOS/LPS [18]. However, PEA transferases are also involved in catalyzing the addition of PEA to HepI of LPS [19,20,21,22,23], O-antigen [24], cellulose [25,26,27], osmoregulated periplasmic glucans [28], and periplasmic proteins [29,30,31] (Figure 2). Considering the evolutionary relationships between different PEA transferases and their substrate specificity (Figure 3 and Table 1), we propose a refined classification of PEA transferases into 11 distinct subclasses. The detailed regulatory framework for each class of PEA transferases is described below.

### 2.1. Class I: Lipid A PEA Transferases

Lipid A PEA transferase or EptA (also called LptA, PmrC, or YjdB in different Gram-negative bacteria), was first identified in *Neisseria meningitidis* [12], and was subsequently shown to be prevalent among clinically relevant Gram-negative bacteria (Table 1 and Figure 2a). In *Neisseria* spp., EptA is the only genomic island-encoded lipid A-modifying enzyme, which transfers PEA to both the 1- and 4′-positions on lipid A to form PPEA-Lipid A; however, the 4′-position is preferable [12]. Notably, the modification of the lipid A headgroups with PEA is specific for neisserial pathogens (*N. meningitidis* and *N. gonorrhoeae*) as the EptA-genomic island is not found in commensal *Neisseria* spp. [50]. In pathogenic *Neisseria*, it was reported that PPEA-Lipid A elicits a stronger cytokine response in THP-1 monocytes than the commensal strains expressing undecorated lipid A [50]. The inactivation of *eptA* in pathogenic *Neisseria*, results in the complete loss of the PEA groups from lipid A and causes the loss of gonococcal colonization of epithelial cells [45]. Similarly, in a pathogenicity study of *N. meningitidis*, the inactivation of EptA resulted in approximately 10-fold reduction in bacterial adhesion to human epithelial and endothelial cells [45]. EptA mutants are more susceptible to killing by human PMNs [51], the complement pathway [47] and cationic antimicrobial peptides (CAMPs), such as PxB (PxB), LL-37 (a human cathelicidin) and mouse cathelicidin, CRAMP-38 [46,47,48]. Additionally, modification of the lipid A moiety has been known to reduce autophagic flux in macrophages and enhance gonoccocal survival during the killing by macrophages [52]. Ultimately, the removal of EptA expression in *N. gonorrhoeae* results in a complete clearance of the bacterium from mice and human models of infection [43].

In *Escherichia coli*, *Salmonella enterica* Typhymurium, and *P. aeruginosa*, however, the modification of lipid A with L-Ara4N residue is more common compared to PEA modification [33,40,100]. EptA can catalyze the addition of PEA residues to both 1- and 4′-phosphate group of Lipid A; however, the reaction at 4′-position only occurs when L-Ara4N is absent [33,40]. L-Ara4N fully neutralizes the negative charge of lipid A, while PEA only decreases it by 0.5. (from −1.5 to 1.0) [33,100], thus conferring an advantage for this hierarchy of lipid A headgroup decoration. Both L-Ara4N and PEA modifications of lipid A decrease the affinity of CAMPs to the bacterial outer membrane, thus promoting CAMP resistance. Intriguingly, *Vibrio cholerae* serogroup O1 consists of two biotypes, Classical and El Tor, but only current pandemic El Tor biotype isolates exhibit CAMP resistance through glycine or diglycine modification of lipid A acyl chain and PEA modification of lipid A headgroups [41,42]. These modifications are catalyzed by AlmEFG and EptA, respectively [41,42]. Previous studies revealed that *eptA* expression in *V*. *cholerae* El Tor only occurs in mildly acidic growth conditions (pH = 5.8) where lipid A glycinylation is absent [41]. Altogether, it is suggested that EptA-mediated modification of lipid A is the secondary mechanism contributing to the CAMP resistance in these pathogens.

In contrast to the previous examples, *H. pylori* PEA transferase decorates only the 1-phosphate headgroup, due to the absence of the 4′-phospate group on the lipid A headgroup [38,39]. Two-enzymatic steps are required for this decoration. Firstly, the 1-phosphate group on the lipid A headgroup is removed by LpxEHP (Hp0021, Lipid A phosphatase), followed by the second step of incorporating the PEA residue by EptA (Hp0022) [38]. In *Haemophilus ducreyi* the addition of PEA to the lipid A headgroups of LOS is required for human defensin resistance but not for cathelicidin LL-37 and human serum resistance, or virulence in a human model of *H. ducreyi* infection [53]. In addition to EptA, the genome of *H. ducreyi* also contains two additional PEA transferases, namely, PtdA and PtdB, which also confer resistance to human defensins, but the exact modification site of these transferases is not clear [53,101].

The addition of PEA to the 4′-phosphate headgroup of lipid A, which is facilitated by ESA_RS09200 in *Cronobacter sakazakii* strain BAA894 (grown at pH 5 but not pH 7), confers resistance to CAMP, decreases recognition by the immune system through TLR4/MD2 cascade signaling, and increases the evasion of phagocytosis [54]. VP_RS21300 is a PEA transferase found in *Vibrio parahaemolyticus* ATCC33846 that selectively modifies lipid A headgroups with PEA, resulting in enhancing pathogenicity in RAW264.7 cells and providing resistance to polymyxins [55]. In *Pasteurella multocida*, the PEA modification of lipid A headgroups is catalyzed by PetL, resulting in CAMP cathelicidin-2 resistance [18]. ICR, chromosome-encoded intrinsic colistin resistance PEA transferase, found in *Moraxella* spp., has been reported to modify the 1 (or 4′)-phosphate position of the lipid A headgroup and contributes to colistin resistance [15,57,102]. In addition to the modification of lipid A moieties, Wei et al. [15] showed that the expression of *Moraxella osloensis* ICR in *E. coli* significantly prevents the generation of reactive oxygen species (ROS) triggered by colistin to kill bacteria.

In 2016, Liu et al. [71] reported the emergence of the first plasmid-mediated colistin resistance gene *mcr-1* in *Enterobacteriaceae*, which was isolated from animals and humans in China. Like EptA, this transmissible PEA transferase mediates 4′-PEA modification of the lipid A headgroups on LPS, conferring polymyxin resistance [71,103]. Shortly afterward, *mcr-2* was identified in colistin-resistant *E. coli* isolated from calves and piglets in Belgium [72]. *mcr-2* has 77% nucleotide sequence identity to *mcr-1* and exhibits high level colistin resistance (4–8 mg/L) [72]. Kieffer et al. [102] showed strong evidence that these transferases potentially originated from *Moraxella* spp. which possess *mcr*-like genes with high nucleotide sequence identity (67–70%) to *mcr-1/2*. In addition, both the *mcr-1/2* and *mcr*-like sites are flanked by the *pap2* gene, a membrane-associated lipid phosphatase [102]. The occurrence of *mcr-3* to *mcr-10* in *Enterobacteriaceae* was reported shortly afterwards [58,59,60,61,62,63,64,65]; however, *mcr-1* is still the most prevalent [104]. The mechanism regulating the expression of *mcr-1* is not well understood, but typically involves regulation by the gene’s promoter as well as activators and/or inhibitors [105]. In a recent study by Yang et al. [106], the authors observed that the over-expression of *mcr-1* confers high-level colistin resistance in *E. coli*; however, it reduced the bacterial fitness and displayed highly attenuated virulence in a *Galleria mellonella* (wax moth) model of infection. Their findings further demonstrated that the expression of *mcr-1* must be strictly controlled in order to maintain both bacterial fitness and colistin resistance [106]. Of note, Yin et al. [107] showed that the expression of *mcr-3* not only confers colistin resistance in *Aeromonas salmonicida* and *E. coli* but also increases pathogenicity and impairs host phagocytosis. In addition to *Enterobacteriaceae, mcr* genes were also identified in *P. aeruginosa* [66,67] and *Acinetobacter* spp. [68,69], respectively. Recently, this family of PEA transferases has spread globally (69 countries), with the potential to create new superbug infections [104].

Cullen et al. [75] first identified *Campylobacter jejuni* EptC (*Cj*EptC), which catalyzes the addition of PEA to 1- and 4′-phosphate groups of lipid A headgroups. In addition to lipid A headgroups, *Cj*EptC is able to catalyze the addition of PEA to other cell-surface residues (Table 1 and Figure 2a,b), including FlgG flagellar rod protein [75], N-linked glycan [76], and the 1st heptose (HepI) of the inner core region of LOS [77]. Deletion of *eptC* in *C. jejuni* showed a 2.4- to 23-fold increase in susceptibility to all tested CAMPs (i.e., PxB, HBD-2, LL-37, P-113, and GAL-6) [77]. Moreover, *C. jejuni* strains lacking *eptC* showed a significant decrease in mobility, flagella production [75], recognition by the TLR4/MD2 complex, and colonization of chick ceca and BALB/cByJ mice, when compared to the WT strains [77], indicating the importance of *eptC* in the pathogenesis. In another study, an insertional mutation of *eptC* significantly decreased biofilm formation (>20%) of *C. jejuni* NCTC11168 on glass and polystyrene surfaces, which are the major food contact surfaces [78].

### 2.2. Class II: PEA Transferases That Modify KdoII of LPS

Kanipes et al. [80] reported the appearance of a KdoII-linked PEA residue on the LPS of a heptose-deficient *E. coli* mutant strain WBB06 when the cells were grown in the medium containing CaCl_2_ (5–50 mm) but not in the standard LB broth. A later study confirmed that EptB (formerly called YhjW) (Table 1 and Figure 2a) is responsible for this modification [81]. The function of EptB-dependent modification of PEA is similar to that of EptA, reducing the negative charge of LPS to develop resistance to PxB [84].

The genome of *S. enterica* Typhimurium also encodes EptB, which specifically catalyzes PEA modification of KdoII in the inner core of LPS [85,86]. A previous study showed that LPS isolated from the *eptB* mutant *S. enterica* Typhimurium but not WT is bound by intelectin, a host protein, which binds to and detoxifies LPS [85]. Consistently, the inactivation of *eptB* was reported to reduce the expression of inflammatory cytokines in the spleen of infected mice [85]. These results suggested the importance of *eptB* in promoting systemic inflammation during infection caused by *S. enterica* Typhimurium.

The *eptB* gene has also been identified in the genome of *Yersinia pestis*, which shares a high sequence identity (63%) and has an absolute functional identity to the *E. coli eptB* [87]. The incorporation of a PEA moiety on Kdo was detected in the LPS of wild-type *Y. pestis* grown at very low temperature (6 °C) while in the same cultivation conditions no PEA was revealed in the LPS of the *eptB* mutant strain [88,89]. Mass spectra analysis of certain *Y. pestis* strains grown at 25 °C also consisted of PEA residue on the LPS [108]. This decoration of PEA is supposed to decrease the susceptibility of *Y. pestis* to the CAMPs produced by the wintering flea [90].

### 2.3. Class III: PEA Transferases That Modify HepI of LPS

In *E. coli*, EptC catalyzes the incorporation of PEA to the Hep I phosphate group of the LPS inner core (Table 1 and Figure 2a) [19]. This modification is thought to increase the sensitivity of bacteria to SDS and sublethal concentrations of Zn^2+^, suggesting that EptC-dependent modification is critical for outer membrane stability [19]. A more recent study found that the PEA moiety incorporated onto the Hep I of LPS contributes to PxB resistance by reducing the OM-permeabilizing activity and binding of PxB to the LPS; however, sensitivity towards sodium dodecyl sulfate (SDS) or novobiocin remained unchanged [20]. Similar to *E. coli*, the homologue CptA (or EptC) identified in *S. enterica* Typhimurium is also required for the addition of PEA to Hep I in the core region of LPS [22]. Interestingly, this modification showed a modest effect on PxB resistance and did not impair virulence in the mouse model [22,23].

### 2.4. Class IV: PEA Transferases That Modify O-Antigen

Lpt-O is a plasmid-encoded PEA transferase found in *Shigella flexneri* that mediates the addition of PEA residues to the 3-position of Rha^II^ on O-antigen (Table 1 and Figure 2a), conferring the MASF IV-1 positive phenotype in serotype Xv strains [24]. The authors hypothesized that this novel mechanism of O-antigen modification may provide the bacterium with a significant fitness advantage, as evidenced by the prevalence of serotype Xv in China [24].

### 2.5. Class V: PEA Transferases That Modify KdoI

In addition to PetL, which is specific for the modification of lipid A headgroups, Harper et al. [18] also identified a PEA transferase PetK in *P. multocida* (Table 1 and Figure 2a). The expression of PetK, which catalyzes PEA modification at KdoI of LPS, is essential for resistance to the CAMP cathelicidin-2 [18].

### 2.6. Class VI: PEA Transferases That Modify Cellulose

Many bacteria are known to produce cellulose, which is a major component of the self-produced extracellular matrix in biofilms [26]. The biosynthesis and export of cellulose in bacteria is typically controlled by the bacterial cellulose synthesis (Bcs) complex (Table 1 and Figure 2b), consisting of the *bcsABZC* operon [109]. Using solid-state NMR, Thongsomboon et al. [26] identified a novel chemically modified cellulose, so-called PEA cellulose, in *E. coli* and *S. enterica*. The modification of cellulose with PEA is catalyzed by BcsG, with the input of cyclic diguanylate monophosphate (c-di-GMP) via a BcsEFG transmembrane signaling pathway [25,26,27]. Eliminating *bcsG* changes the biofilm’s architecture and integrity [25,26,27].

### 2.7. Class VII: PEA Transferases Modifying O-6 Position of HepII of the LPS/LOS

A study conducted using some *N. meningitidis* (meningococci) strains possessing PEA at the O-6 position of HepII identified another transferase, named Lpt6 (Table 1 and Figure 2a), which is encoded by *lpt6*, located on an exchangeable island next to *lgtG*, a galactosyl-transferase [91,110]. Both Lpt6 and LgtG compete for the same attachment site, O-6, on L,D-Hep II. Ram et al. [111] indicated that the binding of complement component C4b (C4b) to the PEA at the O-6 position of HepII was more efficient than to the PEA at the O-3 position, and thereby meningococcal strains harboring PEA-linked to the O-6 position were more sensitive to complement-mediated killing. The high prevalence of meningococcal strains with PEA modification observed at the O-3 position of HepII compared to the O-6 position could be explained by their ability to avoid the clearance of serum via the classical complement pathway [91,97,111]. However, there is no existing evidence in which Lpt6 contributes to CAMP resistance in both *N. meningitidis* and *N. gonorrhoeae* [91,93]. Previous structural studies revealed that *H. influenzae* strains always possess PEA on HepII of LPS [91,94]. Consistent with this, the Lpt6 homologue, which has 75.5% similarity to *N. meningitidis* Lpt6, has been identified in all of *H. influenzae* isolates [91].

The *eptC* gene identified in the genome of *Proteus mirabilis* encodes a PEA transferase EptC, which also involves the transferring of PEA to the O-6 position of L,D-Hep II [95]. In *P. mirabilis*, alterations of LPS with both PEA and L-Ara4N are required for intrinsic resistance to polymyxins [96]. However, some *P. mirabilis* strains isolated from animals do not require EptC for PxB resistance [112]. This observation suggests that L-Ara4N-linked LPS could be the main mechanism of PxB resistance.

### 2.8. Class VIII: PEA Transferases That Modify the O-3 Position of HepII of LOS

Lpt3 was first identified in *N. meningitidis* by Mackinnon et al. [97] and is responsible for the addition of PEA to the O-3 position of the HepII residue of the LOS (Table 1 and Figure 2a). Interestingly, the incorporation of PEA into the inner core of LOS is required for the interaction of the monoclonal antibody B5 to a surface-accessible epitope [97,98]. Inactivation of *lpt-3* resulted in abolishing mAb B5 reactivity and thus decreased the susceptibility of *N. meningitidis* to killing and opsonophagocytosis of mAb B5 [97]. Lpt3 in *N. gonorrhoeae* FA1090 also mediates the modification of the O-3 position of HepII by PEA [99]. Insertional inactivation of *lpt3* did not influence the susceptibility of gonococci to PxB [93,99]. Homologues of *lpt3* have been identified in other Gram-negative bacteria such as *P. multocida* [56]. This gene is also phase variable and responsible for the serotyping of *P. multocida* [56].

### 2.9. Class IX: PEA Transferases That Modify Osmoregulated Periplasmic Glucans

A PEA transferase, namely OpgE, has been identified in *E. coli*, which transfers a PEA moiety to osmoregulated periplasmic glucans (Table 1 and Figure 2b) [28]. However, the role of this modification remains unclear.

### 2.10. Class X: PEA Transferases That Modify Gal Residues of LPS

PetG identified in *P. multocida* has been shown to modify Gal residues of the LPS with PEA (Table 1 and Figure 2a) [18]. The role of this modification remains unclear; however, it is not required for CAMP resistance [18].

### 2.11. Class XI: PEA Transferases That Modify the Pilin Subunit PilE

Pilin phosphorylcholine transferase A (PptA) was first found in *N. meningitidis* and plays a role in catalyzing the attachment of phosphorylcholine to the pilin (Table 1) [29]. Subsequently, this transferase was reported in *N. gonorrhoeae* and can modify the PilE protein subunit of the type IV pilus with both phosphorylcholine and PEA (Figure 2b) [30,31]. Although the mechanism regulating the expression of *pptA* is not yet understood, the PptA-dependent modifications are thought to influence pilin structure and antigenicity [30,31]. It is thought that PptA has a broader substrate range as it has been shown to decorate at least two other periplasmic lipoproteins [113].

## 3. Regulation of PEA Transferase Expression in Pathogenic Gram-Negative Bacteria

In most Gram-negative bacteria, the expression of Class I *eptA* genes is directly regulated by the sensor kinase PmrA/B two-component system in response to multiple environmental signals: For example, Fe^3+^, Al^3+^, or mild acidic pH (Figure 4) [14,23,33,35]. Also, the PhoP/Q sensor kinase two-component system, which is highly conserved among Gram-negative bacteria, indirectly regulates the expression of *eptA* in response to divalent cations (i.e., Mg^2+^, Ca^2+^, low Ph, and CAMPs) [14,23,33,37]. In *P. aeruginosa*, *eptA* is induced by a zinc signal via the ColR/S two-component system [40]. Transcriptomic analysis of *P. multocida* showed that the global regulator Fis and Hfq-dependent sRNA regulate the expression of *petL* both positively and negatively [18]. The expression of *eptA* in *N. meningitidis* and *N. gonorrhoeae* is regulated by stochastic expansion and contraction of a homopolymeric tract in the open reading frame and positively by the MisR/S two-component system, the signal for which remains unknown [114,115,116,117].

**Figure 4 antibiotics-12-01382-f004:**
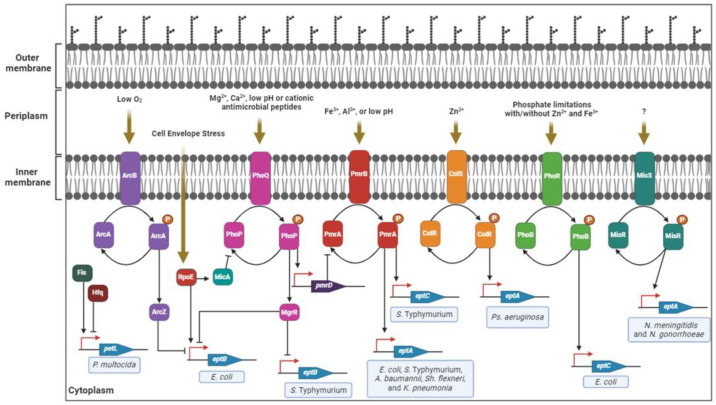
Regulatory pathways to control the expression of PEA transferases in pathogenic Gram-negative bacteria. In several Gram-negative bacteria, the expression of *eptA* is regulated by the two-component systems PhoP/Q and PmrA/B in response to multiple environmental signals [14,23,33,35]. In *P. aeruginosa*, the expression of *eptA* is regulated by a zinc signal via the ColR/S two-component system [40]. The expression of *petL* in *P. multocida* is positively and negatively regulated by the global regulator Fis and Hfq-dependent sRNA [18]. The expression of *eptA* in *N. meningitidis* and *N. gonorrhoeae* is positively regulated by the MisR/S two-component system [114,115,116,117]. The expression of *eptB* is regulated by the sRNA ArcZ via ArcA/B two-component system or sigma factor RpoE and/or sRNA MgrR [82,83]. Two-component systems PmrA/B and PhoB/R regulate the expression of *S. enterica* Typhymurium *eptC* and *E. coli eptC*, respectively [19,22,83]. This figure was created using Biorender.

The expression of Class II *eptB* in *E. coli* requires a complex regulatory system [82] (Figure 4). In brief, the transcription of *eptB* is positively and negatively regulated by the RNA polymerase sigma factor RpoE and sRNA MgrR, respectively [82,83]. In a *phoP*^+^ strain, high concentrations of Ca^2+^ represses MgrR, and allows for the expression of EptB under conditions inducing RpoE expression [82]. Another level of regulation is controlled by ArcZ, a sRNA that negatively regulates *eptB* in a two-component system ArcA/B-dependent manner in response to a low concentration of oxygen [82,83]. In *S. enterica* Typhimurium, the expression of *eptB* is negatively regulated by the sRNA, MgrR [86].

The expression of Class III *eptC* in *E. coli* is positively regulated by the PhoB/R two-component system in response to phosphate-limiting growth conditions with or without Zn^2+^ and Fe^3+^ [19,83]. Interestingly, the expression of EptC was also observed at basal levels without a requirement for regulation [19,83]. Unlike *E. coli*, *S. enterica* Typhimurium *eptC* is positively regulated by the PmrA/B two-component regulatory system [22].

Further studies are required to elucidate the regulatory networks of the remaining classes of PEA transferases.

## 4. Structure of PEA Transferases

### 4.1. Overall Structure

PEA transferases are integral membrane proteins comprised of an N-terminal 5′-helix transmembrane domain and a periplasm-facing C-terminal catalytic domain. Recently, the full-length structure of *N. meningitidis* EptA (*Nm*EptA) was successfully determined (PDB: 5FGN; Figure 5a) [13]. The N-terminal domain of *Nm*EptA has 5 transmembrane α-helices (TMH1-TMH5) positioned approximately parallel to each other, spanning the inner membrane [13]. Two periplasmic facing helices (PH2 and PH2′) are revealed on the long loop, which links between TMH3 and TMH4 [13]. These helices contain both non-polar residues oriented towards the membrane and polar residues oriented towards the periplasm that have been proposed to create the barriers to the substrate binding site of PEA donor lipid [13]. Of these 5 helices, TMH5 is the longest and has adequate length (35 Å) to span the full width of a membrane bilayer (30 Å) while the other four appear shorter than 30 Å [13].

A bridging helix and an extended periplasmic loop coordinate the transmembrane domain with the catalytic domain. Two amphipathic helices (PH3 and PH4) were found on the extended periplasmic loop that adopts a 3_10_ configuration [13]. The presence of positively charged residues (Lys142, Lys144, Arg146, and Lys150) at the end of TMH5 close to the cytoplasmic facing surface is assumed to contribute to the interactions with the phospholipid head groups at the inner membrane [13]. Structural modelling of MCR-1, which used the full-length *Nm*EptA as a template, indicated that the structure of the transmembrane domain of MCR-1 is similar to that of *Nm*EptA with an overall positive charge [118]. Several patches of positively charged R/K amino acids found in the transmembrane region of MCR-1 are proposed to interact with the negatively charged phospholipid head groups of the membrane bilayers [118]. These interactions are required for the stabilization and correct orientation of MCR-1 on the inner membrane, which is essential for catalytic activity [119,120]. Another study revealed that the removal of the TM domain of MCR-2 fully impaired the ability of this enzyme to confer colistin resistance [120].

The crystal structure of the catalytic domain of multiple PEA transferases from Classes I, III and VI, have been solved and they adopt a similar alkaline phosphatase family α/β/α fold, composed of a central seven-stranded β-sheet enveloped by two layers of α-helices (Figure 5b).

**Figure 5 antibiotics-12-01382-f005:**
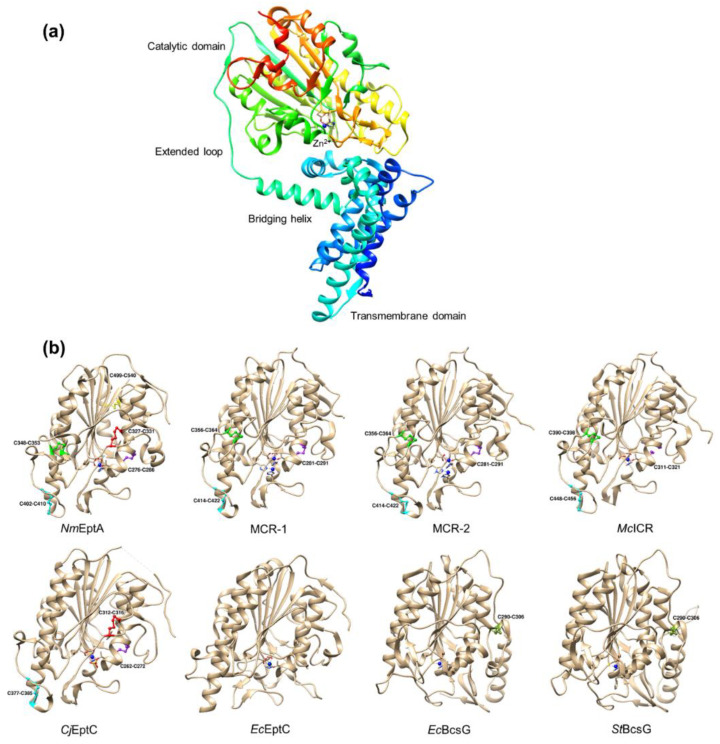
Crystal structure of PEA transferases. Panel (**a**). The full-length crystal structure of *Neisseria meningitidis* EptA (*Nm*EptA). The *Nm*EptA structure (PDB: 5FGN) consists of an N-terminal 5′-helix transmembrane domain and a periplasm-facing C-terminal catalytic domain, which are coordinated by a bridging helix and an extended periplasmic loop [13]. Panel (**b**). Comparison of the disulfide bonds identified in the catalytic domain of PEA transferases. The catalytic domain of *Nm*EptA (PDB: 5FGN) contains five disulfide bonds (DB 1–5), including Cys276– Cys286 (DB1; Purple), Cys327–Cys331 (DB2; Red), Cys348–Cys353 (DB3; Green), Cys402–Cys410 (DB4; Cyan), and Cys499–Cys540 (DB5; Yellow) [13]. The catalytic domain of MCR-1 (PDB: 5GRR) [121] and MCR-2 (PDB: 5MX9) [122] contain three conserved disulfide bonds, including DB1, DB3, and DB4. The catalytic domain of *Mc*ICR (PDB: 6BNF) has two conserved bonds (DB3 and DB4) and one reduced bond (DB1) [57]. *Cj*EptC (PDB: 4TN0) has three conserved bonds (DB1, DB2, and DB4) compared to those of *Nm*EptA [79]. *Ec*EptC (PDB: 6A83) lacks disulfide bonds [21]. *Ec*BcsG (PDB: 6PCZ) [25] and *St*BcsG (PDB: 5OLT) [27] contain a distinct disulfide bond (Cys290–Cys306). This figure was visualized using UCSF Chimera [123].

### 4.2. Disulfide Bonds

The soluble catalytic domains of all classes of PEA-transferases contain variable numbers of disulfide bonds (DB) and these have been shown to have roles in structural integrity and active site architecture. The soluble catalytic domain of the Class I *Nm*EptA (PDB: 5FGN) contains five pairs of cysteine residues forming five disulfide bonds (DB 1-5), namely, Cys276–Cys286 (DB1), Cys327–Cys331 (DB2), Cys348–Cys353 (DB3), Cys402–Cys410 (DB4), and Cys499–Cys540 (DB5) (Figure 5b) [13,49]. Multiple sequence alignment analysis showed that only DB1 and DB4 are structurally conserved across all Class I PEA transferases acting on lipid A headgroups on LPS/LOS [13]. DB1 bridges a smaller α-helix to the central β-strand whereas DB4 stabilizes a loop located at the top of the enzyme [121]. The other three DBs have a variety of proposed roles. In the MCR proteins, DB3 may structurally constrain the conformational freedom of the loop (aa: Lys348–365) and facilitate the entry of the substrate into the catalytic cleft [121]. The absence of this extended loop in *Cj*EptC may allow the entry of multiple substrates that are consistent with the ability of *Cj*EptC to modify different targets [79]. The roles of DB2 and DB5 are less clear but are most likely to be involved in protecting the enzyme from proteolytic degradation by periplasmic proteases [21,124].

DB distribution in other classes of PEA transferase is different from Class I such as in *Ec*EptC (Class III), which lacks disulfide bonds [19,21]. Class VI *Ec*BcsG and *St*BcsG structures contain one distinct DB (Cys290–Cys306), which connects the loop just after the corresponding helix with an adjacent helical component positioned alongside the central β-sheet structure of the catalytic domain [25,27]. Anderson et al. [25] showed that mutation of this Cys pair to alanine in *Ec*BcsG resulted in inactivation, suggesting a functional role. Overall, the presence of DBs across PEA transferases is believed to be necessary for structural integrity and catalytic activity.

### 4.3. Active Site Architecture

How the active site works was established using the canonical crystal structure of Class I *Nm*EptA [13]. The active site of *Nm*EptA lies at the base of a cleft formed by the extensive interface between the catalytic and transmembrane domains, while its substrate binding pocket involves the catalytic domain and the PH2 and PH2′ helices on the transmembrane domain [13]. The active site structure of *Nm*EptA (Figure 6) revealed a bound Zn^2+^ ion and a phosphorylated form of threonine, Thr280, which is required for the nucleophilic attack of the substrate, phosphatidylethanolamine (PE) (Figure 1), and the formation of an intermediate in which PEA is linked to Thr [33]. The side chains of His453, Asp452, Glu240, and nucleophile residue Thr280 tetrahedrally coordinate this Zn^2+^ ion and are conserved among the other PEA transferases [49]. An additional Zn^2+^ ion, which is coordinated by His383, His465, and an oxygen atom of the phosphate group attached to Thr280, was found when the protein crystal was soaked in a solution containing ZnSO_4_ [49]. The organization and architecture of the active sites of other Class I PEA transferases MCR-1/2, *Mc*ICR, and *Cj*EptC, all contain the same conserved set of coordinating residues with a putative nucleophilic threonine and Zn^2+^ ions; however, the number of Zn^2+^ ions is varied (Figure 6) [13,57,70,79,118,121,122,125,126,127]. Point mutations of these active-site residues result in loss of function of the enzyme [21,79].

The structure of *Ec*EptC (Class III) revealed a conserved Thr288 in the active site, which was not phosphorylated and may represent the native state of the enzyme without any bound substrate [21]. Although the Zn^2+^-coordinating residues of *Ec*EptC are conserved in other PEA transferases, only two residues (Glu245 and Asp461) are essential for zinc co-ordination [21].

Both catalytic domains of *Ec*BcsG [25] and *St*BcsG [27] (Class VI) revealed the presence of a Zn^2+^ ion, which is coordinated by the conserved side chain residues Cys243, Glu442, and His443. Of note, serine but not threonine is predicted to be a catalytic nucleophile residue in both structures [25,27]. Compared to the reported structures of Class I PEA transferases, the second-metal binding site of *Ec*BcsG and *St*BcsG contains only a conserved His396 residue while another, His458, is substituted by Arg458. Consistently, no sign of a second Zn^2+^ ion was detected in electron density maps [27]. Interestingly, the replacement of Arg458 by alanine or methionine but not histidine resulted in the severe impairment of cellulose synthesis of *S. enterica* Typhimurium strain MAE97, demonstrating that Arg458 is involved in the substrate recognition and/or catalysis rather than zinc coordination [27]. The Tyr277 residue in the active site pocket is predicted to facilitate the integration of cellulose glycans adjacent to catalytic Ser278 and Zn^2+^ ion at the active site [25]. Anderson et al. [25] showed that substitution of the Zn^2+^-binding residues Cys243, Glu442, and His443, catalytic nucleophile Ser278, His396, and Tyr277 of *Ec*BcsG with alanine resulted in failure to maintain the fragile pellicle phenotype of *E. coli* strain AR3110, highlighting the importance of these residues for enzymatic activity in vivo. In *S. enterica* Typhimurium, Sun et al. [27] observed that the point mutations at Zn^2+^-binding residues, Ser278, and His396 resulted in *St*BcsG protein variants with disturbed cellulose biosynthesis confirming the importance of these residues in transferase activity.

### 4.4. Model of Enzyme Activity

Previous studies revealed that the catalytic domain of Class I PEA transferases, such as *Nm*EptA and MCR enzymes, can adopt multiple conformational states to accommodate two different-sized substrates (PE and lipid A) [13,70,125,128,129,130]. This intriguing observation suggests the involvement of a possible ping-pong mechanism in transferring PEA from PE donor to lipid A, necessitating two distinct reaction steps (Figure 7) [70,125,128,129,130].

In the first step, the membrane phospholipid PE enters the active site pocket, where a Zn^2+^ ion coordinates with the phosphoryl group of PEA [70,125,128,129]. Subsequently, the PEA moiety from the PE donor is directly transferred to the nucleophile residue Thr285 [70,125,128,129]. Density functional theory cluster model calculations suggested that this transition state involves the concerted cleavage of P–O bonds with the transfer of two protons, one from Thr285 to the carboxylate of Glu246 and the second from His395 to the dephosphorylated lipidic leaving group [70,125,129]. This step requires a Zn^2+^ ion to stabilize the nucleophilic state of Thr285, leading to the nucleophilic attack of Thr285 on PE, breaking the acyl side chain from PE and releasing diacylglycerol [129] (Figure 1). Using an in vitro enzymatic assay, several studies have shown that purified PEA transferases can cleave the PEA group from a fluorescently labelled substrate NBD-Glycerol-3-PEA and release NBD-Glycerol [128,130,131,132]. This step is believed to be common among PEA transferases due to their conserved core structures and the same cavity for PE entry [30,128].

The second step occurs when the receiver substrate lipid A binds to the enzyme, enabling the transfer of the PEA moiety to the 1- and/or 4′-phosphate group to generate the final products PPEA-1 (and/or 4′)-lipid A (Figure 7) [128,129]. The calculations of Suardiaz et al. [129] on MCR-1 suggested that the presence of the second Zn^2+^ ion is mandatory for this step of the reaction as it coordinates the oxygen atoms of the phosphate group of lipid A and then guides it to the phosphate group of PEA, which is attached to Thr285 [129]. A proton from Glu246, which was protonated in the first step of the reaction, is returned to Thr280, which subsequently breaks the P-O bond between Thr285 and PEA moiety and restores the coordination of Thr285 [129]. The His466 residue now connects to the second Zn^2+^ ion, which may enable the release of the modified lipid A [129]. The enzyme finally restores to its resting state where the second Zn^2+^ ion is released and the coordination of His466 to the first Zn^2+^ ion is restored [129]. The second step of the reaction is supposed to be common in Class I PEA transferases recognizing the same substrate lipid A, but it may differ in other PEA transferases with different receiver substrates [21,25,27,49,57,70,79,122].

### 4.5. Co-Crystals with Receiver Substrates

Currently, there is very little understanding of how different receiver substrates are bound to the enzyme-PEA intermediate. Most of our current knowledge stems from studies on Class I enzymes *Nm*EptA [13] and *Mc*ICR [57] with analogues of PE and lipid A. Since lipid A is highly flexible and hydrophobic, no complete co-crystal structure has been reported. Anandan et al. [13] did report a bound dodecyl maltoside (DDM) detergent molecule, which is a PE substrate analogue, in the substrate pocket of *Nm*EptA (PDB: 5FGN; Figure 8a), when the protein was solubilized and purified in DDM. The O3B hydroxyl group of DMM has been shown to bind to an active site Zn^2+^ and the hydroxyl group of Thr280 while the carbohydrate moiety of DDM forms hydrogen bonds with three invariant residues in the PH2 and PH2′ helices. Xu et al. [128] conducted a molecular docking analysis of *Nm*EptA with DDM and identified a substrate entry/binding cavity composed of 12 residues. Apart from the five residues that coordinated the zinc molecule, seven residues (Asn106, Thr110, Glu114, Ser325, Lys328, His383, and Asp465) were purportedly involved in the recognition of substrate PE [128]. Point mutation of Glu114Ala, Lys328Ala, His383Ala, and Asp465Ala led to loss of enzyme function, while N106A, E114A, and S325A resulted in partial inactivation as determined by polymyxin resistance [128].

In a separate study, the *Mc*ICR^Thr315Ala^ mono-Zn^2+^ structure in complex with PEA (Figure 8b) by Stogios et al. [57] showed that the phosphate portion of the PEA moiety forms connections with the Zn^2+^, His511, Ala315, and three water molecules. The structure also featured the requirement of the dimerization of the *Mc*ICR catalytic domain for the coordination of the substrate that provides additional interaction with Tyr338 from the partner subunit [57]. Interestingly, no interactions between the ethanolamine group of PEA and the enzyme were observed [57]. The attempt to determine the crystal structure of *Mc*ICR^Thr315Ala^ in complex with the lipid A headgroup was unsuccessful due to the absence of density corresponding to the substrate molecule [57].

Glucose and xylose are analogues of the glucosamine backbone of lipid A. A solved structure of the co-crystal of the catalytic domain of MCR-1 in complex with D-glucose (Figure 8c) identified a binding pocket composed of Thr283, Ser284, Tyr287, Pro481, and Asn482 residues the phosphorylated Thr285 in the active site [118]. Mutation of the Tyr287 and Pro481 suggested that they are significant for functionality [118]. D-xylose (Figure 8d) is bound to the same pocket with a weaker interaction than that of D-glucose [133].

## 5. Progress in Drug Development of PEA Transferase Inhibitors

There are many programs underway to identify Class I lipid A PEA transferase inhibitors. Lipid A PEA transferase inhibitors can be used to break resistance to clinically used antibiotics such as polymyxin and colistin. These programs generally fall into three categories: The development of analogues of the donor or receiver molecules, inhibition by interference with Zn^2+^ occupancy in the active site, or screens for compounds with affinity for EptA that result in inhibition of the enzyme.

### 5.1. Inhibitors Based on Analogues

Wei et al. [118] showed that ethanolamine, a PE substrate analogue, bound to the cavity pocket of MCR-1 and impaired PxB resistance, but it had low activity (10 mM) against an *E. coli* strain expressing *mcr-1* (Table 2). A solved structure of the co-crystal of ethanolamine with the catalytic domain of MCR-1 [118] (Figure 8e) depicted a possible substrate-binding pocket consisting of eight residues (Glu246, Thr285, Asn329, Lys333, His395, Asp465, His466, and His478). Notably, all point mutations of these residues impaired PxB resistance, implying that they are crucial for the catalytic activity of MCR-1 [118]. Lan et al. [134] later developed more effective derivatives of 1-phenyl-2-(phenylamino) ethenone, 6p [4-((1-Ethoxy-2-(4-hexylphenyl)-2-oxoethyl)amino)-3-methylbenzoic Acid], and 6q [4-((1-Ethoxy-2-(4-octylphenyl)-2-oxoethyl)amino)benzoic Acid)] that completely reversed colistin resistance in *E. coli* strain BL21(DE3) carrying *mcr-1*. Results from an enzymatic assay revealed that the compounds inhibited the PEA transfer reaction catalyzed by MCR-1 [134]. Molecular docking studies further revealed that the most potent compounds formed hydrogen bonds with the cavity pocket residues Glu246 and Thr285 [134].

### 5.2. Targeting Zinc Co-Ordination in the Active Site of EptA

Targeting the Zn^2+^ ion, which is necessary for enzymatic activity, is a promising strategy for developing inhibitors of PEA transferases. Despite having distinct structures and substrates, both MCRs and metallo-β-lactamases can be effectively disrupted by auranofin, an antirheumatic drug, which displaces Zn^2+^ in the active sites of both enzymes [135]. Screening of a library of metal-based compounds, bioligands, and Zn^2+^ chelating agents identified silver nitrate (AgNO_3_), which restores the colistin activity against multidrug-resistant *Enterobacteriaceae* isolates carrying *mcr-1* and *mcr* variants [136]. X-ray crystallographic analysis revealed that silver disrupts the function of MCR-1 by displacing the zinc ion, thus interfering with substrate binding to the enzyme [136].

PBT2, a zinc ionophore originally developed for the treatment of Alzheimer’s disease, has been found to possess antimicrobial properties against various Gram-positive bacteria by interfering with bacterial metal homeostasis [137]. Studies by Jen et al. [138,139] revealed that the combination of PBT2 and zinc enhances the effectiveness of tetracycline, colistin, PxB, and LL-37 against *N. gonorrhoeae* strains. Although the PBT2/zinc combination has been shown to partially inhibit the decoration of lipid A with PEA in WHO Z, the precise mechanism of action is not yet fully understood [138].

According to Cui et al. [140], the combination of EDTA and Baicalin (a compound from *Scutellaria baicalensis* (a herbal Chinese medicine)) could restore colistin efficacy against several colistin-resistant *Salmonella* isolates recovered from poultry and humans by directly inhibiting the production of MCR-1. Molecular docking analysis revealed a strong interaction between baicalin and several residues of MCR-1 (Lys307, Lys333, Asp331, Asp337, Gly334, and Tyr308), whereas EDTA, a strong metal chelating agent, was proposed to chelate the Zn^2+^ in the active site of MCR-1 [140].

### 5.3. Miscellaneous Compounds with No Known Mode of Action

Numerous studies have performed non-targeted screening programs for the identification adjuvants that break resistance to colistin. Most of these compounds have no known direct mode of action targeting MCR-1. Several promising candidates, including Pterostilbene [141], Genistein [142], Phloretin [143], Osthole [144], Pogostone [145], Pingwei Pill [146], Pyrazolones [147], and Honokiol [148], which could improve polymyxin efficacy both in vitro and in vivo, have been found. Molecular simulation and/or molecular docking analyses suggest that most of these compounds, except for Pterostilbene, have the potential to interact with multiple residues in the MCR-1 active site [141,142,143,144,145,146,148], but further studies are necessary to confirm their direct ability to inhibit MCR-1.

**Table 2 antibiotics-12-01382-t002:** Inhibitors acting on lipid A PEA transferases.

Inhibitor	Structure	Target	Organism	Working Concentration	Activity	Target Verification	Reference
Ethanolamine	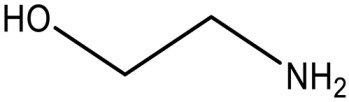	MCR-1	*E. coli*	10 mM	Complete reversal of PxB resistance	The crystal structure of ethanolamine binding to MCR-1.	[118]
Compounds 6p	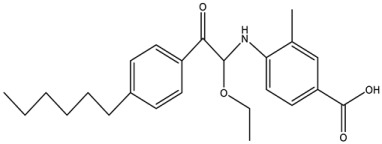	MCR-1	*E. coli*	25 μM	Complete reversal of colistin resistance	Inhibit the reaction of PEA transfer catalyzed by MCR-1 in an enzymatic assay.	[134]
Compounds 6q	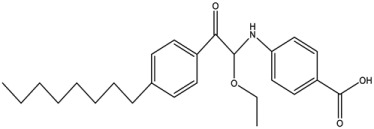	MCR-1	*E. coli*	25 μM	Complete reversal of colistin resistance	Inhibit the reaction of PEA transfer catalyzed by MCR-1 in an enzymatic assay.	[134]
Auranofin	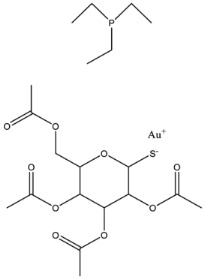	MCR-1 and MCR variants	*E. coli* and other G-ve species *	7.37 μM	Complete/partial reversal colistin resistance	Inhibit the reaction of PEA transfer catalyzed by MCR-1 in an enzymatic assay. Displace Zn^2+^ by Au^+^ ion in a zinc release assay (PAR assay) and X-ray crystallography.	[135]
Silver nitrate	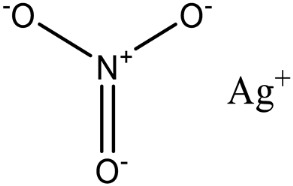	MCR-1 and MCR variants	*E. coli* and other G-ve species **	2.96 μM	Complete/partial reversal of colistin resistance	Inhibit the reaction of PEA transfer catalyzed by MCR-1 in an enzymatic assay. Displace Zn^2+^ by Ag^+^ ion in a zinc release assay (PAR assay) and X-ray crystallography.	[136]
EDTA/Baicalin ^#^	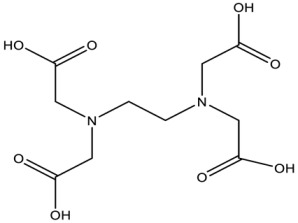	MCR-1	*Salmonella* isolates	EDTA: 213.86 μM/Baicalin: 2800.41 μM	Complete reversal of colistin resistance	Directly inhibit *mcr-1* expression.	[140]
PBT2/zinc ^##^	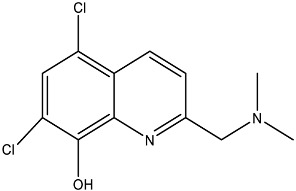	EptA	*N. gonorrhoeae*	PBT2: 0.5 μM/zinc: 2.5 μM	All tested strains became sensitive to tetracycline, colistin, PxB, LL-37, and PG-1.	Reduce PEA decoration of lipid A by MS analysis.	[138,139]
Compound 137	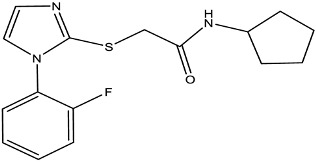	EptA	*N. gonorrhoeae*	100 μM	Reduced PxB MIC by 4-fold for FA1090 and 2-fold for WHO reference strains.	Reduce PEA decoration of lipid A by MS analysis.	[149]
Compound 2	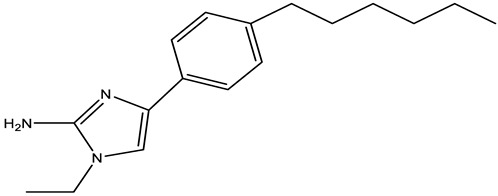	MCR-1	*A. baumannii*, *E. coli*, *K. pneumoniae*	30 μM	Reversed colistin resistance mediated by MCR-1 and chromosome-encoded enzymes.	Reduce PEA decoration of lipid A by MS analysis.	[150]
IMD-0354	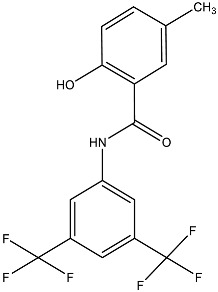	Lipid A modifying enzymes	*A. baumannii*, *E. coli*, *K. pneumoniae*, *P. aeruginosa*	5 μM	Reversed colistin resistance mediated by MCR-1 and chromosome-encoded enzymes.	Reduce or completely abrogate PEA and Ara4N modification of lipid A by MS analysis.	[151]

Note: The table only includes the compounds that have been verified to inhibit the activity of lipid A PEA transferases. * *S. flexneri*, *S. enterica* Typhimurium*, K. pneumoniae*, *Enterobacter asburiae*, *Klebsiella aerogenes*, *Enterobacter kobei;* ** *S. enterica* Typhimurium, *K. aerogenes*, *K. pneumoniae*, *E. kobei;*
^#^ Structure of EDTA (Ethylenediaminetetraacetic acid); ^##^ Structure of PBT2; MS: Mass spectrometry; ITC: Isothermal titration calorimetry; PxB: Polymyxin B; MCR variants: MCR-1.5, MCR-1.6, MCR-1.7, MCR-1.9, MCR-1.10, MCR-2.1, MCR-3.1, MCR-4.1, MCR-5.1, MCR-6.1, MCR-7.1; PAR assay: 4-(2-pyridylazo)resorcinol assay.

### 5.4. Rational Drug Development Programs

Three drug screening programs have been conducted to identify inhibitors of lipid A PEA transferases [149,150,151]. These inhibitors do not directly impact bacterial growth; instead, they enhance the susceptibility of bacteria to CAMPs. Mullally et al. [149] were the first to utilize saturation transfer difference (STD)-NMR screening, coupled with iterative medicinal chemistry optimization, to create the first generation of EptA inhibitors against *N. gonorrhoeae*. The lead compound 137 (Table 2) improved the susceptibility of strain FA1090 and multiple MDR strains to PxB by 4-fold and 2-fold, respectively. Pre-treating bacterial cells with the compound resulted in increased macrophage-mediated killing. Moreover, compound treatment of bacteria reduced lipid A decoration with PEA by 17%, and decreased cytokine responses in THP-1 cells [149].

Barker et al. [150] identified a compound (compound 2), which suppresses *mcr-1* encoded colistin resistance in *A. baumannii*, *E. coli*, and *K. pneumoniae*. Exposure of *A. baumannii* ATCC 17978*^+mcr−1^* to the compound resulted in a reduction in the extent of lipid A modification, as demonstrated by mass spectrometry analysis of the lipid A [150].

The screening of kinase inhibitor libraries identified IMD-0354, which successfully eliminated colistin resistance mediated by Lipid A modification in several Gram-negative organisms without affecting growth [151]. Analysis of the lipid A extracted from *A. baumannii* 4106 and *K. pneumoniae* B9, treated with IMD-0354 using mass spectrometry, indicated that the presence of the compound could partially or completely inhibit PEA and Ara4N decoration of lipid A [151]. Although the targets of IMD-0354 are not completely known, it could potentially target the regulatory pathways, which control PEA and Ara4N decoration of lipid A. Further optimization of IMD-0354 analogues led to another compound, which restored the susceptibility of a highly resistant strain of *K. pneumoniae* to colistin at 250 nM; nevertheless, further study is required to validate its target on lipid-A modifying enzymes [152].

## 6. Concluding Remarks and Future Perspectives

The lipid A PEA transferases present across all clinically relevant Gram-negative bacteria that confer resistance to CAMPs, are considered to be the most important subclass of PEA transferases. In contrast to other Gram-negative bacteria, pathogenic *Neisseria*, including *N. gonorrhoeae* and *N. meningitidis*, utilize EptA as the sole mechanism to modify lipid A headgroups with PEA. This unique feature highlights EptA as a potential therapeutic target, particularly for the management of MDR gonorrhoea.

The increasing prevalence of mobilized *mcr* genes encoding PEA transferases is a growing public health concern as it raises the possibility of global dissemination of polymyxin resistance. Accordingly, current drug development efforts for lipid A modifying PEA transferases have been directed towards MCR-1, with the goal of restoring polymyxin sensitivity, but these endeavors are still in the preclinical stage.

Given the conserved structure of PEA transferases, additional investigations are needed to evaluate the feasibility of developing broad-spectrum inhibitors against multiple Class I enzymes, which could have significant clinical benefits. Further research also needs to focus on determining the full-length enzyme-substrate co-crystal structure, which could help to design potent and specific inhibitors for these enzymes.

## Figures and Tables

**Figure 1 antibiotics-12-01382-f001:**
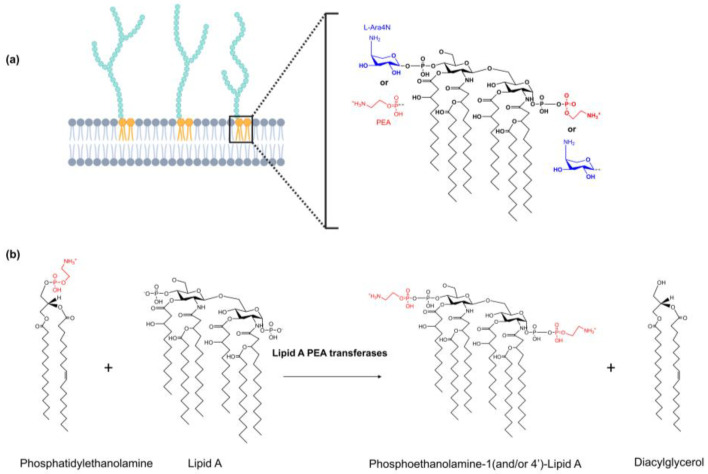
Modification of lipid A headgroups on LPS/LOS in pathogenic Gram-negative bacteria. Panel A. The modification of lipid A headgroups on LPS/LOS with phosphoethanolamine (PEA) and 4-amino-4-deoxy-L-arabinose (L-Ara4N) confer resistance to cationic antimicrobial peptides (CAMPs) in pathogenic Gram-negative bacteria (**a**). Panel B. The lipid A PEA transferases catalyze the reaction in which the PEA moiety is transferred from phosphatidylethanolamine (PE) onto the 1 and/or 4′ position of the Lipid A headgroups on LPS/LOS (**b**). This figure was created using Biorender.

**Figure 2 antibiotics-12-01382-f002:**
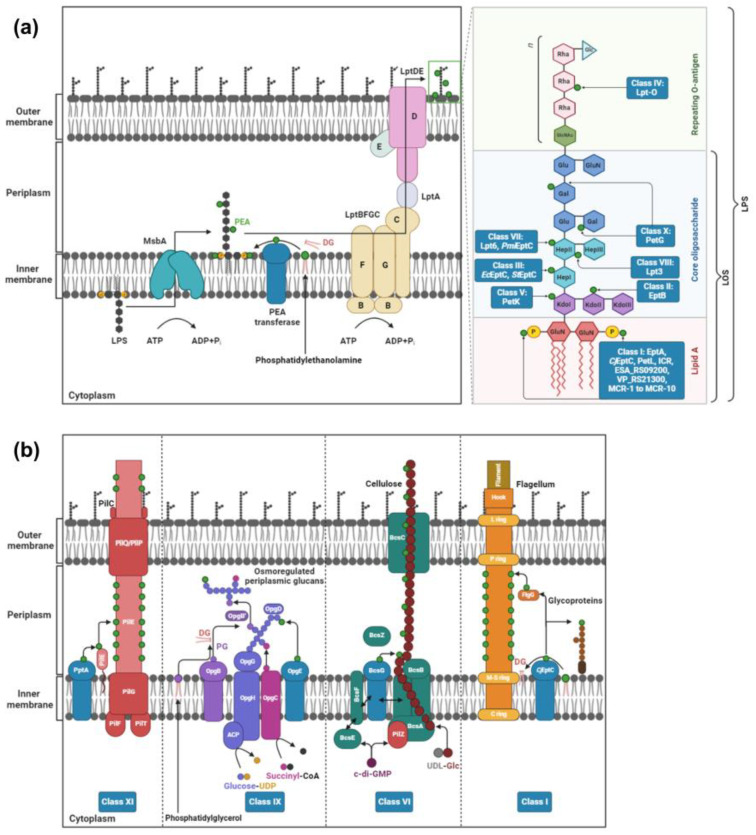
Schematic models of the PEA modifications catalyzed by 11 subclasses of PEA transferases in pathogenic Gram-negative bacteria. Panel (**a**) PEA transferases modify lipooligosaccharide (LOS)/lipopolysaccharide (LPS) during transport across the periplasmic space. Multiple positions on LOS/LPS can be decorated (see insert showing Classes I, II, III, IV, V, VII, VIII, X). Panel (**b**) Other targets in the periplasm are decorated by Classes I, VI, IX and XI. Class I *Cj*EptC (*Campylobacter jejuni* EptC) modifies lipid A headgroups of LOS, HepI of LOS, FlgG flagellar rod protein, and glycoproteins. Class VI BcsG modifies cellulose [26]. Class IX OpgE modifies osmoregulated periplasmic glycans [28]. Class XI PptA modifies Pilin subunit PilE [29]. This figure was created using Biorender.

**Figure 3 antibiotics-12-01382-f003:**
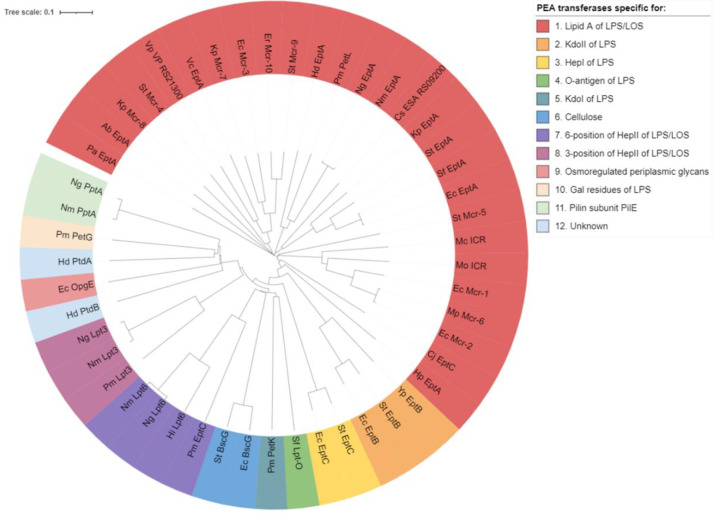
Maximum likelihood phylogeny inferred from amino acid sequences of PEA transferases. Tree representing the evolutionary relationships between different PEA transferases were created after multiple sequence alignment using the Clustal Omega web server (https://www.ebi.ac.uk/Tools/msa/clustalo/, accessed on 12 July 2023). The tree was further modified using the iTOL web server (https://itol.embl.de/, accessed on 12 July 2023) [32], with 11 PEA transferase subclasses identified based on their sites of PEA modification. Detailed representative PEA transferase sequences used to construct this tree are provided in Appendix A. *Escherichia coli* (Ec), *Salmonella enterica* Typhimurium (St), *Acinetobacter baumannii* (Ab), *Shigella flexneri* (Sf), *Klebsiella pneumoniae* (Kp), *Helicobacter pylori* (Hp), *Pseudomonas aeruginosa* (Pa), *Vibrio cholerae* (Vc), *Neisseria meningitidis* (Nm), *Neisseria gonorrhoeae* (Ng), *Haemophilus ducreyi* (Hd), *Cronobacter sakazakii* (Cs), *Vibrio parahaemolyticus* (Vp), *Proteus mirabilis* (Pmi*), Pasteurella multocida* (Pmu), *Moraxella catarrhalis* (Mc), *Moraxella osloensis* (Mo), *Moraxella pluranimalium* (Mp), *Enterobacter roggenkampii* (Er), *Yersinia pestis* (Yp), *Campylobacter jejuni* (Cj), and *Haemophilus influenzae* (Hi).

**Figure 6 antibiotics-12-01382-f006:**
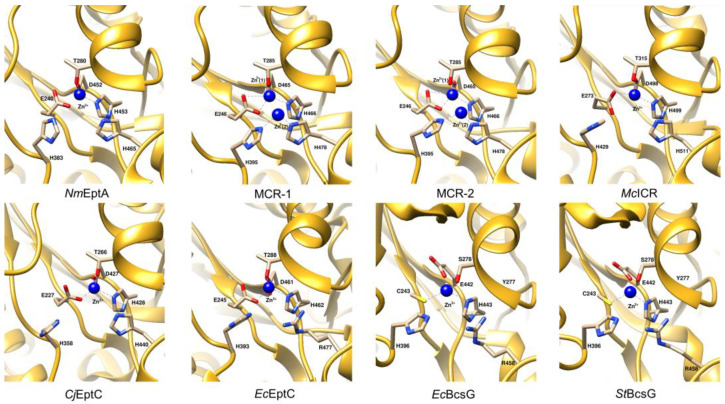
Comparison of the active sites of PEA transferases. *Nm*EptA (PDB: 5FGN) [13], MCR-2 (PDB: 5MX9) [122], *Mc*ICR (PDB: 6BNF) [57], *Cj*EptC (PDB: 4TN0) [79], *Ec*EptC (PDB: 6A83) [21], *Ec*BcsG (PDB: 6PCZ) [25], and *St*BcsG (PDB: 5OLT) [27]. The structure of MCR-1 (PDB: 5GRR) [121], which displays two zinc molecules at the active site region, was chosen for the comparative analysis. This figure was visualized using UCSF Chimera [123].

**Figure 7 antibiotics-12-01382-f007:**
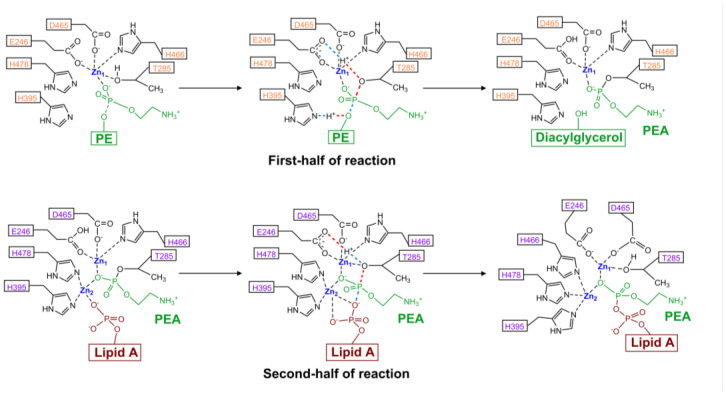
Reaction catalyzed by Class I PEA transferases. A cluster model has been proposed to illustrate the catalytic pathway of MCR-1. The first half of the reaction involves the formation of a PEA-Enzyme conjugate and the release of diacylglycerol. In the second half of the reaction the PEA-lipid A once formed will be released from the Enzyme active site. Amino acids are numbered according to their position in MCR-1 and side chains are shown. Dashes represent hydrogen bonds while solid lines are covalent bonds. The figure was modified from Suardiaz et al. [129].

**Figure 8 antibiotics-12-01382-f008:**
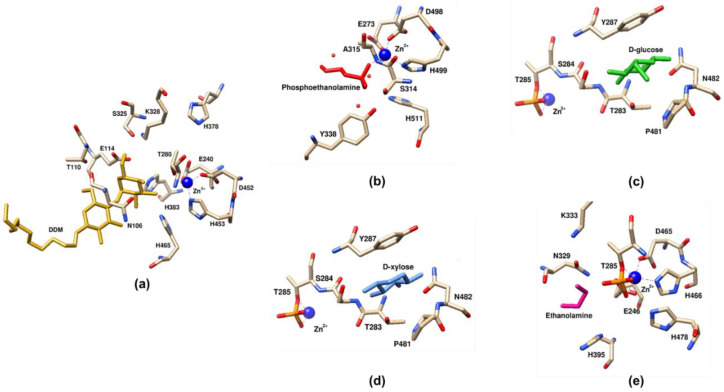
The interactions between PEA transferases and their corresponding substrate or substrate analogues. Panel (**a**) depicts the interaction between *Nm*EptA and DDM molecule (PDB: 5FGN) [13]. Panel (**b**) shows the interaction between the *Mc*ICR^Thr315Ala^ mono-Zn^2+^ catalytic domain and phosphoethanolamine (PDB: 6BND) [57], which involves Tyr338 from the partner subunit. Red sphere: Water molecule. Panels (**c**–**e**) illustrate the interactions between the catalytic domain of MCR-1 and D-glucose (PDB: 5YLF) [118], D-xylose (PDB: 5ZJV) [133], and ethanolamine (PDB: 5YLE) [118], respectively. This figure was visualized using UCSF Chimera [123].

**Table 1 antibiotics-12-01382-t001:** Summary of known PEA transferases identified in pathogenic Gram-negative bacteria.

Class	Enzyme	Organism	Encoded by	Sites of PEA Modification	Regulation	Function	Reference
I	EptA (LptA, PmrC, YjdB)	*E. coli*	Core genome	Lipid A headgroups	PmrA/B and PhoP/Q	Resistance to CAMPs	[17,23,33]
		*S. enterica* Typhymurium	Core genome	Lipid A headgroups	PmrA/B and PhoP/Q	Resistance to CAMPs	[17,23,33,34]
		*A. baumannii*	Core genome	Lipid A headgroups	PmrA/B	Resistance to CAMPs	[35,36]
		*S. flexneri*	Core genome	Lipid A headgroups	PmrA/B and PhoP/Q	Resistance to CAMPs	[17]
		*K. pneumonia*	Core genome	Lipid A headgroups	PmrA/B and PhoP/Q	Resistance to CAMPs	[17,37]
		*H. pylori*	Core genome	Lipid A headgroups	Requires the removal of phosphate group by LpxEHP (Hp0021, Lipid A phosphatase)	Resistance to CAMPs	[38,39]
		*P. aeruginosa*	Core genome	Lipid A headgroups	ColR/S	Resistance to CAMPs	[17,40]
		*V. cholera*	Core genome	Lipid A headgroups	pH = 5.8	Resistance to CAMPs	[41,42]
		*N. meningitidis*	Genetic island	Lipid A headgroups	Phase variation of the open reading frame and MisR/S	Resistance to CAMPs Resistance to human serum Colonisation of mucosal surfaces	[12,17,43,44,45,46,47,48,49]
		*N. gonorrhoeae*	Genetic island	Lipid A headgroups	Phase variation of the open reading frame and MisR/S	Resistance to CAMPs Resistance to human serum Evasion of phagocytosis by PNMs and macrophages Colonisation of mucosal surfaces Influence bacterial survival in mouse and human models	[12,17,43,44,45,46,47,48,50,51,52]
		*H. ducreyi*	Core genome	Lipid A headgroups	Unknown	Resistance to CAMPs	[53]
	ESA_RS09200 (ESA_02008)	*C. sakazakii*	Core genome	Lipid A headgroups	pH = 5.0	Resistance to CAMPs Avoid TLR4/MD2 recognition Evasion of phagocytosis by macrophages	[54]
	VP_RS21300	*V. parahaemolyticus*	Core genome	Lipid A headgroups	pH = 6.5	Resistance to CAMPs Increase pathogenicity	[55]
	PetL	*P. multocida*	Core genome	Lipid A headgroups	Global regulator Fis and Hfq-dependent sRNA	Resistance to CAMPs	[18,56]
	ICR	*Moraxella* spp. *(M. catarrhalis* and *M. osloensis*)	Core genome	Lipid A headgroups	Unknown	Intrinsic resistance to CAMPs	[15,57]
	MCR-1 to MCR-10	*Enterobacteriaceae*, *Moraxella* spp., *P. aeruginosa* and *Acinetobacter* spp.	Plasmid	Lipid A headgroups	Unknown	Resistance to CAMPs Mcr-3 increases pathogenicity and impairs phagocytosis.	[58,59,60,61,62,63,64,65,66,67,68,69,70,71,72,73,74]
	EptC	*C. jejuni*	Core genome	Lipid A headgroups HepI of LOS FlgG flagellar rod protein N-linked glycan (Glycoproteins)	Unknown	Resistance to CAMPs Mobility and flagella production Biofilm formation Recognition by a human TLR4/MD2 Colonization of chick ceca and BALB/cByJ mice	[75,76,77,78,79]
II	EptB	*E. coli*	Core genome	KdoII of LPS	MgrR/RpoE (High Ca^2+^ concentration) ArcZ sRNA in an ArcA/B-dependent manner (Low oxygen concentration)	Resistance to CAMPs	[80,81,82,83,84]
		*S. enterica* Typhymurium	Core genome	KdoII of LPS	MgrR	Resistance to CAMPs Protect bacteria from the binding and detoxifying by intelectin Enhance cytokine secretion in spleen of infected mouse	[85,86]
		*Y. pestis*	Core genome	KdoII of LPS	Low temperatures (6 °C)	Resistance to CAMPs	[87,88,89,90]
III	EptC (CptA)	*E. coli*	Core genome	HepI of LPS	PhoB/R	Resistance to SDS and sublethal concentration of Zn^2+^ Not confer resistance to CAMPs	[19,20,21]
		*S. enterica* Typhymurium	Core genome	HepI of LPS	PmrA/B	Modest resistance to CAMPs but not impairing the virulence in mouse model of infection.	[22,23]
IV	Lpt-O	*S. flexneri*	Plasmid	O-antigen of LPS	Unknown	Confer the MASF IV-1 positive phenotype in serotype Xv strains	[24]
V	PetK	*P. multocida*	Core genome	KdoI of LPS	Unknown	Resistance to CAMPs	[18]
VI	BcsG	*E. coli*	Core genome	Cellulose	Unknown	Cellulose production Maintain biofilm’s architecture and integrity	[25,26]
		*S. enterica* Typhymurium	Core genome	Cellulose	Unknown	Cellulose production Maintain biofilm’s architecture and integrity	[27]
VII	Lpt6	*N. meningitidis*	Genetic island	O-6 position of HepII of LOS	Unknown	Increase susceptibility to complement-mediated killing	[91,92]
		*N. gonorrhoeae*	Genetic island	O-6 position of HepII of LOS	Unknown	Unknown	[93]
		*H. influenzae*	Core genome	O-6 position of HepII of LPS	Unknown	Unknown	[91,94]
	EptC	*P. mirabilis*	Core genome	O-6 position of HepII of LPS	Unknown	Resistance to CAMPs	[95,96]
VIII	Lpt3	*N. meningitidis*	Core genome	O-3 position of HepII of LOS	Unknown	Require for the binding of mAb B5	[92,97,98]
		*N. gonorrhoeae*	Core genome	O-3 position of HepII of LOS	Unknown	Unknown	[99]
		*P. multocida*	Core genome	O-3 position of HepII of LPS	Unknown	Inhibit the binding of mAbs T1C6 and T6B2	[56]
IX	OpgE	*E. coli*	Core genome	Osmoregulated periplasmic glycans	Unknown	Unknown	[28]
X	PetG	*P. multocida*	Core genome	Gal residues of LPS	Unknown	Unknown	[18]
XI	PptA	*N. meningitidis*	Core genome	Pilin subunit PilE	Unknown	Influence pilin structure and antigenicity	[29]
		*N. gonorrhoeae*	Core genome	Pilin subunit PilE	Unknown	Influence pilin structure and antigenicity	[30,31]
N/A	PtdA	*H. ducreyi*	Core genome	Unknown	Unknown	Resistance to CAMPs	[53]
N/A	PtdB	*H. ducreyi*	Core genome	Unknown	Unknown	Resistance to CAMPs	[53]

Footnote: CAMPs: Cationic antimicrobial peptides; PNMs: Polymorphonuclear neutrophils; TLR4/MD2: Toll-like receptor 4-myeloid differentiation factor 2 complex; SDS: Sodium dodecyl sulfate; MASF: Monoclonal antisera of *S. flexneri*; mAb: Monoclonal antibody; N/A: Not available due to lacking site of PEA modification. For full species designations refer to the legend of Figure 2.

## Data Availability

Not applicable.

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
