# Peer review of "Phosphoethanolamine Transferases as Drug Discovery Targets for Therapeutic Treatment of Multi-Drug Resistant Pathogenic Gram-Negative Bacteria"

_antibiotics, 2023, doi:10.3390/antibiotics12091382_

Round 1

Reviewer 1 Report

The manuscript entitled "Phosphoethanolamine transferases as drug discovery targets 2 for therapeutic treatment of multi-drug resistant pathogenic 3 Gram-negative bacteria" provides a comprehensive overview of the function, structure, and mechanism of action of the phosphoethanolamine transferases identified in pathogenic Gram-negative bacteria. The manuscript is written well and is of great interest to the scientific community working in the area of drug discovery & development for antimicrobial resistance. 

Suggestions:

1. Include the chemical structures of inhibitors presented in Table 2. 

2. Selection of words in some places is inappropriate. Rectify. 

Selection of words in some places is inappropriate. Rectify. 

Author Response

Thank you reviewer 1 for your comments.

We have added the chemical structures to Table 2 and fixed the errors in the text throughout.

Reviewer 2 Report

The paper titled "Phosphoethanolamine transferases as drug discovery targets for therapeutic treatment of multi-drug resistant pathogenic Gram-negative bacteria" discusses the growing issue of antibiotic resistance caused by multi-drug resistant (MDR) bacteria. Specifically, the review focuses on the role of phosphoethanolamine (PEA) transferases in Gram-negative bacteria and their potential as drug targets for tackling antibiotic resistance.

Polymyxins are considered as the last resort antibiotics for treating MDR Gram-negative bacterial infections. However, the emergence of resistance to polymyxins poses a significant challenge in empirical therapy. Bacteria employ various mechanisms to protect themselves against polymyxins, and one of the main mechanisms is the modification of the lipid A of the outer membrane by adding PEA moieties to the lipid A headgroups. The review provides a comprehensive overview of the function, structure, and mechanism of action of PEA transferases identified in pathogenic Gram-negative bacteria. These enzymes not only modify the lipid A but also decorate proteins and glycans, further contributing to bacterial resistance.

The review highlights the significance of targeting PEA transferases for drug development to combat antibiotic resistance. By inhibiting these enzymes, it may be possible to reverse the resistance and restore the efficacy of polymyxins in empirical therapy. The paper discusses various PEA transferases, including EptA, EptB, EptC, Lpt3, Lpt6, PetG, PptA, and BcsG, and their involvement in the modification of different bacterial components. Furthermore, the paper summarizes the progress made in drug development targeting PEA transferases. It also emphasizes the importance of understanding the structure-function relationship of these enzymes for the successful design of potential therapeutic agents.

In conclusion, the review sheds light on the crucial role of PEA transferases in antibiotic resistance in Gram-negative bacteria. It provides valuable insights into their function, mechanism of action, and their potential as drug targets. Targeting these enzymes could open up new possibilities for the development of effective treatments against MDR Gram-negative bacterial infections, ultimately addressing the global public health challenge of antibiotic resistance.

Author Response

Reviewer 2- thank you for your positive response to our reviw.

Reviewer 3 Report

This is a well-written review of phosphoethanolamine transferases, which form an important family of enzymes related to antibiotic resistance.  Consequently, interest in the manuscript will be wide-spread, since resistance is such a hot topic.  The English is clear, and the paper well organized.  The Figures are clearly presented, and the references are extensive.

My only scientific issue relates to the compound structures mentioned in Section 5.  The text refers to compounds 6p and 6q, for example, on line 599.  These numbers are meaningless unless one refers to the original paper, so this is not informative.  I strongly recommend including at least structures rather than compound numbers.  Some compounds like "auranofin" (line 609) can at least be googled, so perhaps these structures are not needed so much.  But as a chemist, I would like to be able to see all the structures easily, for example to look for possible patterns in structure-activity relationships.

One other minor point: section 2.4 is entirely in italics.  Presumably only the title should be italicized.

Overall, I think if these points are addressed, then the paper will be a useful contribution and should be published.

Author Response

Thank you reviewer 3 for your kind comments on the review. We have added the chemical structures to Table 2 and fixed the italics formatting issue.